



# Determining an optimal transport velocity in the marginal ice zone using operational ice-ocean prediction systems

Graig Sutherland[1], Victor Aguiar[2,3], Lars-Robert Hole[2], Jean Rabault[4,5], Mohammed Dabboor[1], and Øyvind Breivik[2,6]

[1]Environmental Numerical Prediction Research, Environment and Climate Change Canada, Dorval, QC, Canada
[2]Norwegian Meteorological Institute, Bergen, Norway
[3]Department of Physics and Technology, The Arctic University of Norway, Tromsø, Norway
[4]Norwegian Meteorological Institute, Oslo, Norway
[5]Department of Mathematics, University of Oslo, Oslo, Norway
[6]Geophysical Institute, University of Bergen, Bergen, Norway

**Correspondence:** Graig Sutherland (graigory.sutherland@ec.gc.ca)

**Abstract.** Knowledge of transport in the marginal ice zone (MIZ) is critical for operations in the Arctic and associated emergency response applications, for example, the transport of pollutants, such as oil, as well as predicting drift associated with search and rescue operations. This paper proposes a general transport equation for the MIZ that can be used for operational purposes in the MIZ. This equation is designed to use a mean velocity of the ice and water velocity, which is weighted by the ice concentration. A key component is the introduction of a leeway coefficient for both the ocean and ice components. These leeway coefficients are determined by minimizing the velocity error between the transport model and observed drifter velocity in the MIZ. These leeway values are found to be 3% of the wind for the water leeway and 2% and 30° to the right of the wind for the ice leeway, which are consistent with "rule of thumb" values for surface drifters and sea ice respectively. This general transport model is compared with other transport models and the error is reduced by a factor of 2 compared with traditional transport models for 48 hour lead times. The inclusion of a leeway coefficient in the ice is the key component to reduce trajectory errors in the MIZ.

## 1 Introduction

Estimating the drift and transport of material in the marginal ice zone (MIZ) is a crucial aspect of polar operations. This can be related to search and rescue operations (Rabatel et al., 2018) as well as the transport of oil and other contaminants (French-McCay et al., 2017; Nordam et al., 2019). The requirement of accurate drift and transport estimates for these processes is



exacerbated by the remoteness of these regions. Thus, accurate predictions up to at least 48 hours are required to provide
sufficient time to coordinate response efforts. Due to the numerous transient processes in the ocean, these short-term predictions
can be particularly difficult to predict accurately (Christensen et al., 2018).

The first step in predicting transport in the MIZ is accurate predictions of ice, water and wind velocities, and in the arctic
there are a few operational products which provide forecasts of at least 48 hours. A couple examples of such Arctic systems
being the Regional Ice-Ocen Prediction System (RIOPS) in Canada (Dupont et al., 2015) and the TOPAZ system as part of
the Copernicus Marine Environment Monitoring Service (CMEMS) (Sakov et al., 2012). These operational prediction systems
typically will have the ocean and ice components coupled, and typically do not include forcing from surface waves. Rarely
they are coupled with the atmosphere, but one example of an ice-ocean-atmosphere coupled model is the Canadian Arctic
Prediction System (CAPS) which is the RIOPS system two-way coupled with the GEM (Global Environmental Multi-scale)
atmospheric model (Côté et al., 1998b, a; Girard et al., 2014). The ice models used in these prediction systems assume a
continuous ice cover and a viscous-plastic rheology, which is typically derived from principles introduced by Hibler III (1979).
For ice concentrations of less than 80%, the ice is assumed to be in "free drift" and the rheology is expected to have a minimal
impact on the dynamics (Hibler III, 1979). While free drift models exist, and they perform quite well when compared with
observations, they do require data in order to tune the drag coefficients (Schweiger and Zhang, 2015) which can be difficult to
obtain in the arctic.

It is common for operational prediction of drifting objects to include a leeway term, which is a fraction of the wind that is
added to the predicted water velocity. This has been common for a long time in search and rescue (Breivik et al., 2011) and oil
spill trajectory modelling (Spaulding, 2017). These leeway values can represent direct wind forcing on the object (Kirwan Jr
et al., 1975), and/or wind-dependent physics which are not included in the operational prediction system. As most operational
marine prediction systems don't include surface waves it is common to include the Stokes drift (Breivik and Christensen, 2020)
which can be approximated by a leeway coefficient (Breivik et al., 2011; Sutherland et al., 2020). In addition, material at the
surface, such as oil but also ice, will strongly attenuate surface waves creating an additional force on the material (Weber,
2001). Oil and ice also impact the "roughness" of the surface; slicks in the case of oil which reduce the roughness (Wu, 1983),
and the presence of form drag in the MIZ which can increase roughness (Lüpkes et al., 2012; Tsamados et al., 2014). For oil,
it's long been established that the competing wind and wave effects combine for a net leeway of about 3% of the velocity (Wu,
1983). For sea ice it is not clear whether a leeway term should be included in the drift, however, operational models such as
CAPS and TOPAZ do not explicitly include processes associated with surface waves or form drag due to the ice roughness.
Therefore, a non-zero leeway term in the MIZ may be necessary.

There are only a few examples of using a leeway coefficient with the drift of sea ice, but these are typically restricted to cases
where there are no ocean current measurements and the ice speed is modelled as solely a function of the wind. Wilkinson and
Wadhams (2003) observed an ice leeway which was dependent on ice concentration. The leeway values ranged from 3.9% for
ice concentrations less than 25% to 2.2% for ice concentrations greater than 75%. More recently, Lund et al. (2018) observed
that the wind and sea ice motion is not always well correlated and that ice leeway coefficients span a large range of values
from 3.5 to 5% across a wide range of ice concentrations in the arctic MIZ. In these papers, it is possible that some of the ice





motion was due to tidal and/or inertial motions, which will not directly scale with the instantaneous wind and would lead to uncertainties in the above estimates.

     While typically leeway coefficients are determined for particular objects using detailed field measurements (Breivik et al., 2011), recently it was shown by Sutherland et al. (2020) that leeway coefficients can also be predicted using operational prediction systems. The mean leeway coefficients using the operational prediction data were found to be consistent with

detailed observations and independent of the choice of operational prediction system. However, these principles have only been applied in open ocean conditions and have not been tested in the presence of sea ice. It would seem that there are certain analogues with the open ocean that would warrant the use of a leeway coefficient in the MIZ. First, most operational ice-ocean models don't include surface waves, which are not uncommon to the MIZ and will impact the drift (Weber, 1987; Squire, 2020). In addition, in the MIZ where ice concentrations are typically less than 80%, the ice is said to be in free drift and the ice

motion will depend strongly on a accurate parameterization of the drag coefficient (Cole et al., 2017) and not on an accurate rheology of the internal ice stress (Hibler III, 1979). Form drag due to sea ice is also another physical process which will effect the ice motion and has a strong effect in the MIZ (Lüpkes et al., 2012).

     In this paper, we compare the observed drift of four ice drifters in the MIZ and compare these with empirical leeway models. These drifters span ice concentrations from 10 to 90%. Inputs for the ocean and ice velocities are obtained from two operational

ice-ocean prediction systems focused on the arctic. The outline of the paper is as follows. Section 2 a leeway model is developed for the MIZ. Section 3 describes the data, the ice-ocean prediction systems and the methodology used in calculating the leeway coefficients. Results are presented in section 4 followed by the conclusions.

## 2   Transport equations for the MIZ

Ice and ocean surface velocities in the MIZ are strongly coupled. In the dynamical models of operational prediction systems,

this is through the relative friction between the ocean and ice components. Most operational ice-ocean prediction systems will have the ice and ocean components two-way coupled. Differences between ice and ocean surface velocities can arise due to ice inertia, which is generally negligible in the MIZ, and internal stresses within the ice, which are also negligible for ice concentrations less than 80% (Hibler III, 1979). So the ice and ocean surface velocities in the MIZ are predominantly in a steady-state free drift mode, with the magnitude, as predicted by operational ice-ocean prediction systems, being dependent on

the drag coefficients and the ice concentration.

     Given the uncertainties associated with modelling velocities in the MIZ, it makes sense to use both ice and ocean velocities to derive a general transport velocity. This is precisely the approach used for oil spill modelling in ice-covered waters (Nordam et al., 2019), where a mean transport velocity is calculated and the ice and surface ocean components are weighted by a function of ice concentration. The model used by the oil spill community uses some empirical estimates for the weighting function as

well as for the leeway. We will use this idea of a mean transport model as a template, but will make it more general for a wider use. First, we will go through the oil transport equations and then develop a more general model for transport in the MIZ.



## 2.1 Oil transport equation in the MIZ

The basic equation for the advection velocity in ice-covered water used by the oil spill community (Nordam et al., 2019) is

$$\mathbf{u}_o = k_i \mathbf{u}_i + (1 - k_i)(\mathbf{u}_w + \alpha_w \mathbf{U}_{10}), \tag{1}$$

where $\mathbf{u}_o$ is the oil velocity, $\mathbf{u}_w$ is the water velocity, $\mathbf{u}_i$ is the ice velocity, $\mathbf{U}_{10}$ is the wind velocity at 10 m, $\alpha_w$ is a leeway coefficient for oil in water which is typically about 3% (Spaulding, 2017; Nordam et al., 2019) and $k_i$ is the ice transfer coefficient and a function of ice concentration $A$. Often $k_i$ is presented as a piece-wise linear function of ice concentration (Nordam et al., 2019), commonly referred to as the "80/30" rule, and defined as

$$k_i^{80/30} = \begin{cases} 0 & \text{if } A < 0.3, \\ \frac{A - 0.3}{0.5} & \text{if } 0.3 \le A < 0.8, \\ 1 & \text{if } 0.8 \le A. \end{cases} \tag{2}$$

The arguments for the "80/30" rule originated from observations by Venkatesh et al. (1990) and are qualitative in nature. Venkatesh et al. (1990) observed that for sea ice concentrations greater than 80% the oil appeared to drift with the surrounding ice. For ice concentrations less than 30%, the oil drifted with the water. In between these two limits a linear weighting is assumed. The functional form of (2) has not been investigated in detail (French-McCay et al., 2017; Nordam et al., 2019), but any form for $k_i$ will inevitably be a monotonically increasing function of ice concentration with the limits $k_i = 0$ when $A = 0$ (no ice) and $k_i = 1$ when $A = 1$ (all ice).

## 2.2 General transport equation in the MIZ

For the transport of material in the MIZ we propose a more general equation than (1) that allows for a non-zero leeway coefficient in the ice,

$$\mathbf{u}_o = k_i (\mathbf{u}_i + \alpha_i \mathbf{U}_{10}) + (1 - k_i)(\mathbf{u}_w + \alpha_w \mathbf{U}_{10}), \tag{3}$$

where $\alpha_i$ is the leeway coefficient in the ice. As mentioned previously, $k_i$ in (3) can be any monotonically increasing function of ice concentration $A$ that has the limits $k_i = 0$ at $A = 0$ and $k_i = 1$ at $A = 1$. The simplest parameterization of $k_i$ that satisfies these conditions is the linear relation $k_i = A$ and this will be used for this study.

The inclusion of a leeway coefficient in the sea ice has certain analogues with the open ocean. First, most operational ice-ocean models don't include surface waves, which are not uncommon to the MIZ and will impact the drift (Weber, 1987; Squire, 2020). There are also advantages to using both the ice and ocean components for transport. One potential advantage is that the wind stress in coupled ice-ocean prediction systems (Sakov et al., 2012; Dupont et al., 2015), as well as for wave models (Rogers et al., 2016), is partitioned as a linear function of ice concentration and, therefore, biases in the ice concentration could lead to biases in the ice and ocean velocities. Using a weighted mean ice-ocean velocity ensures that all of the wind-generated currents will be present and biases can be adjusted via the leeway coefficient. Essentially, the assumption is that we



will formulate a transport model which assumes that a weighted average of the ice and ocean velocities in the MIZ, along with a corresponding leeway coefficient to be determined, will provide a more accurate estimate for transport velocities than using either the ocean or the ice velocities separately. In some ways this is similar to how wave models calculate their solutions, that is they take a weighted mean of the forcing functions, weighted by ice concentration, and calculate one wave spectrum for both the ice and water components combined, even though on the sub-grid scale the waves are most likely different under ice than

under no ice.

## 3  Data and Methodology

### 3.1  Drifters

Four drifters, designed to measure wave-ice interactions (Rabault et al., 2020), were deployed on various ice floes in the MIZ approximately 250 km north of Svalbard on 19 September 2018. The initial location for each drifter was chosen to sample a

broad range of ice conditions ranging from solitary floes (approximately 10% ice coverage) to more densely packed sea ice (approximately 90% ice coverage) in a transect perpendicular to the ice edge (Figure 1).

The drifters are equipped with an inertial motion unit to measure the directional wave spectra and a GPS sensor which provides accurate measurements of the geographic location and time. The data are recorded on each drifter and sent via Iridium approximately every 3 hours. The drifter velocities were calculated using the forward difference in geographic locations.

The drifters transmitted data from 19 September 2018. Two drifters survived until 1 October 2018 (14435 and 14437) while the other two (14432 and 14438) stopped reporting at approximately 26 September 2018. From the available data it appears that each of the drifters stopped transmitting when they had left the MIZ (Figure 1).

### 3.2  Ice-ocean prediction systems

The Canadian Arctic Prediction System (CAPS) is a coupled atmosphere-ice-ocean prediction system, with separate analyses

for the atmosphere, ice and ocean components, which is run operationally at Environment and Climate Change Canada (ECCC) as part of the Year of Polar Prediction. The ice-ocean component is a 1/12° coupled NEMO-CICE configuration as in Dupont et al. (2015). The ice-ocean component is coupled with an atmospheric model on a higher (3 km) grid. The dynamical core of the atmospheric component of CAPS is GEM (Global Environmental Multiscale), a non-hydrostatic model which solves the fully compressible Euler equations (Côté et al., 1998b, a; Girard et al., 2014).

TOPAZ is a coupled ice-ocean data assimilation system covering the North Atlantic and Arctic oceans (Sakov et al., 2012). TOPAZ represents the Arctic component of the Copernicus Marine Environment Monitoring Service (CMEMS) system (marine.copernicus.eu). Atmospheric forcing is provided by the European Centre for Medium-Range Weather Forecasting (ECMWF). The ocean component of TOPAZ is the HYbrid COordinate Model (HYCOM), which is a hybrid model consisting of $z$-level vertical coordinates in the upper (mixed) layer and isopycnal coordinates below. The horizontal resolution is about





**Figure 1.** Drifter tracks (-) and locations (o) at three different times during their respective trajectories. Contours of ice concentration are shown as calculated by CAPS (a, c, e), and TOPAZ (b, d, f). Drifter IDs, from furthest in the ice to nearest to ocean, are 14432 (blue), 14437 (orange), 14435 (green) and 14438 (red). Drifters exit the MIZ at approximately 2018-09-25 00:00 UTC.

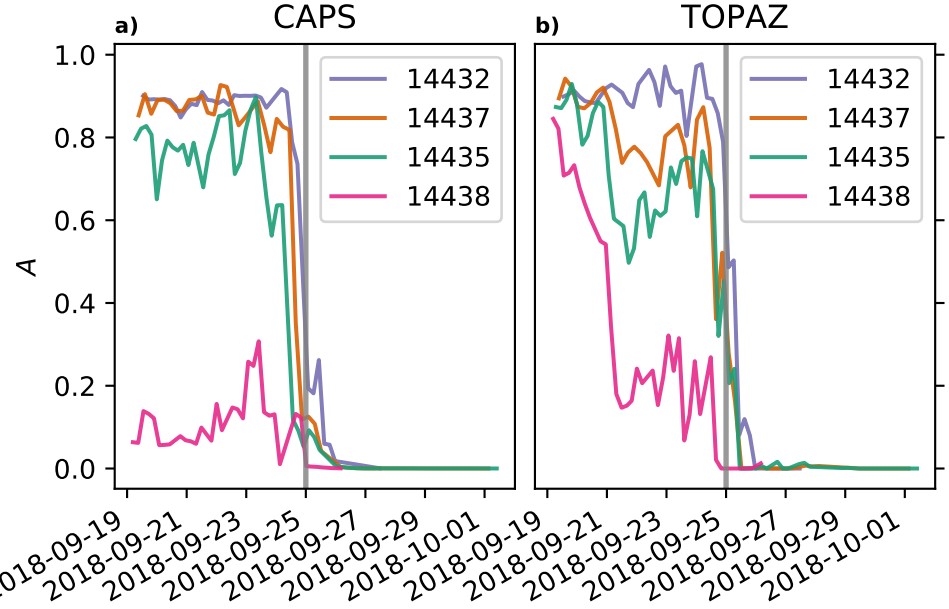

**Figure 2.** Ice concentration interpolated to drifter tracks as calculated by a) CAPS and b) TOPAZ. The vertical grey line shows the approximate time (2018-09-25 00:00 UTC) that the drifters leave the MIZ.

12 km in the Arctic. The ocean model is coupled with a single thickness category ice model with elastic-viscous-plastic (EVP) rheology. Details can be found in Sakov et al. (2012) and Xie et al. (2017).

For all simulations, the wind forcing from CAPS is used. The CAPS model has a grid resolution of roughly 3 km, includes data assimilation, and is freely available. TOPAZ is forced by ECMWF IFS forecast data which has a spatial resolution of $0.1°$ ($\simeq 9$ km) on a regular lat/lon grid (https://resources.marine.copernicus.eu/documents/PUM/CMEMS-ARC-PUM-002-
ALL.pdf). The choice of using only one of the wind forcing data sets is partially motivated by the ECMWF IFS forecast data requiring a license, but also allows for a simpler analysis to focus on the ice-ocean systems.

Snapshots of the ice concentration field (Figure 1) as well as the along-track time series of ice concentration (Figure 2) are calculated for both CAPS and TOPAZ. A consistent feature between the two ice concentrations is that after approximately 25 September 2018 both CAPS and TOPAZ predict that there is no ice at the drifter locations. This will allow us to test the model
of (3) for an object that travels between ice and open water. There are also clearly some differences between the predicted ice concentrations, which are most likely due to the different assimilation cycles of each model. While the analysis of the ice field in CAPS is updated for each forecast (every 12 hours) the TOPAZ fields have a weekly analysis cycle, which could lead to the differences, especially for the two drifters which begin in lower ice concentrations (drifters 14435 and 14438).

A comparison of the model and observed velocities for each drifter is shown in Figure 3. Time series for ice, water and wind
(2%) velocities, interpolated in time and space to available drifter observations, are shown. It is apparent from Figure 3 that the observed drifter velocities are nearly always greater than either of the ice and water velocities from CAPS or TOPAZ. Ice



and ocean velocities in CAPS vary in magnitude and direction more than those from TOPAZ. This is most likely due to ocean tides being included in CAPS, while these are not in TOPAZ, as this region is known to have large diurnal tidal currents due to a near-resonant, forced topographic wave propagating around the Yermak Plateau (Hunkins, 1986; Padman et al., 1992).

### 3.3 Calculating leeway coefficients

As the drifters were deployed on ice floes, it is assumed that the drift of these ice floes in the MIZ can be given by (3). Ice drifters have been previously used by French-McCay et al. (2017); Babaei and Watson (2020) in comparing transport equations with the drift of ice floes. However, even for ice floes over a wide range of ice concentrations, the weighting of (3) states that the ice floe will drift more like the water in low ice concentrations and more like the ice in high ice concentrations. The primary assumption for using the ice floes as a proxy for general transport is that the ice floe is small enough that inertial forces are negligible. To investigate this assumption, the results from using (3) will be compared with results using (1) as well as an ocean-leeway model ($k_i = 0$, $\alpha_w = 0.03$) and an ice-only model ($k_i = 1$, $\alpha_i = 0$).

The optimal leeway coefficients to be used in (3) are determined by minimizing the mean absolute error (MAE) between the observed and predicted velocities. The leeway coefficients are assumed to be constant in time, and are a vector with both a downwind and crosswind component. It is also important to assess the sensitivity to these leeway coefficients, which is not trivial as the calculated MAE is a function of four variables as each $\alpha_i$ and $\alpha_w$ are vectors. To look at the sensitivity, 2-D slices are made through the 4-D MAE field to show the relative sensitivity for $\alpha_i$ for a fixed $\alpha_w$ and vice versa for $\alpha_w$ for a fixed $\alpha_i$. The fixed points for these 2-D slices are selected from averaging the optimal leeway values for all the drifters and ice-ocean prediction systems.

## 4 Results and discussion

### 4.1 Ice and water leeway coefficients

The MAE between the observed drifter velocity and the model velocity from (3) is minimized for each drifter and choice of ice-ocean prediction system. The optimal leeway coefficients, as well as the MAE in km/day, are located in Table 1 for using the CAPS forcing and Table 2 for the TOPAZ forcing. To demonstrate the relative sensitivity to the choice of leeway coefficients, 2-D slices of the MAE contours are made through a fixed leeway coefficient. That is, the downwind and crosswind contours for the ice leeway are presented for a fixed water leeway and vice versa. Optimal leeway values are found to be approximately $\alpha_w = 0.03$ and $\alpha_i = 0.02e^{-i\pi/6}$ and these are used for the constant leeway coefficients used for the MAE slices. No leeway angle is used for $\alpha_w$, but for $\alpha_i$ an angle of -30° is found where the negative implies clockwise rotation relative to the wind as we are using complex notation for our vectors. The MAE for these fixed leeway coefficients are also shown in Tables 1 and 2, denoted MAE* in each Table, and the difference is less than 1.5 km/day for each. Although not shown here, the MAE contours are qualitatively similar for different choices of fixed leeway coefficients with the primary difference being in the magnitude of the MAE.



**Figure 3.** Magnitude and direction of observed drifter velocities ($\mathbf{u}_o$, black) and predicted wind velocity ($2\%\mathbf{U}_{10}$, orange) ice ($\mathbf{u}_i$) and water velocities ($\mathbf{u}_w$) from CAPS (blue) and TOPAZ (green) interpolated to the drifter locations. The strong diurnal tidal current is clearly observed in CAPS, but is absent from TOPAZ as this model does not include tides. The vertical grey line shows the approximate time when the drifters exit the MIZ.



**Figure 4.** MAE contours (in km/day) between observed drift velocities with (3) for the along and cross-wind components of $\alpha_i$ through the constant $\alpha_w = 0.03$. The left column uses the CAPS forcing and the right column uses TOPAZ forcing. The black dot shows the location of the MAE minimum and the black contour line shows the MAE value within 1 km/day of the minimum. Each row is for an individual drifter in order from high ice concentration at the top to low ice concentration at the bottom. Sensitivity to to the choice of $\alpha_i$ is much greater in the high ice concentration than the low.

**Table 1.** Best fit for leeway coefficients using the CAPS forcing. MAE$^*$ is the mean absolute error using leeway values of $\alpha_i$ = 2% and Ang($\alpha_i$) = -30° and $\alpha_w$ = 3%. Negative angles imply clockwise direction relative to wind.

| Drifter | $|\alpha_i|$ | Ang($\alpha_i$) | $|\alpha_w|$ | Ang($\alpha_w$) | MAE (km/day) | MAE$^*$ (km/day) |
|---------|------|------|------|------|------|------|
| 14432 | 1.7% | -30° | 2.3% | -5° | 9.4 | 10.0 |
| 14437 | 2.1% | -25° | 2.9% | 5° | 11.3 | 11.4 |
| 14435 | 2.4% | -35° | 2.5% | 5° | 11.9 | 12.3 |
| 14438 | 2.8% | -45° | 2.7% | 5° | 13.4 | 13.5 |

**Table 2.** Best fit for leeway coefficients using the TOPAZ forcing and constant . MAE$^*$ is the mean absolute error using leeway values of $\alpha_i$ = 2% and Ang($\alpha_i$) = -30° and $\alpha_w$ = 3%. Negative angles imply clockwise direction relative to wind.

| Drifter | $|\alpha_i|$ | Ang($\alpha_i$) | $|\alpha_w|$ | Ang($\alpha_w$) | MAE (km/day) | MAE$^*$ (km/day) |
|---------|------|------|------|------|------|------|
| 14432 | 2.9% | -25° | 3.3% | -30° | 9.8 | 12.3 |
| 14437 | 3.2% | -25° | 2.8% | -5° | 11.8 | 13.0 |
| 14435 | 3.4% | -30° | 2.7% | -10° | 10.6 | 12.1 |
| 14438 | 1.5% | -5° | 2.9% | -15° | 11.5 | 12.2 |

Contours of MAE for $\alpha_i$ for the downwind and crosswind components at a fixed value of $\alpha_w$ are shown in Figure 4. The MAE contours for each drifter are very similar when using the CAPS data or the TOPAZ data. There is a clear sensitivity related to the ice concentration as the drifter furthest in the ice (14432) has the sharpest contour gradients in MAE and the drifter at the ice edge (14438) shows very little change in MAE for different values of $\alpha_i$. Not including drifter 14438, the optimal value for $\alpha_i$ is approximately 0.02 and 30° to the right of the wind using the CAPS data and 0.03 and 30° to the right of the wind using the TOPAZ data. The difference between 0.02 and 0.03 are within the 1 km/day MAE contour, which is less than 10% of the total MAE.

Contours of MAE for $\alpha_w$ for the downwind and crosswind components at a fixed value of $\alpha_i$ are shown in Figure 5. Not surprisingly, the MAE contours in Figure 5 have the opposite sensitivity as those in Figure 4, with sharper contour gradients in MAE for the lowest ice concentration and MAE showing minimal sensitivity to the choice of $\alpha_w$ in high ice concentration. Also, similar to Figure 4, the MAE values in Figure 5 are similar in magnitude when using either CAPS or TOPAZ data and range between 10 and 14 km/day.

## 4.2 Short-term trajectory predictions

Lagrangian trajectories are simulated with MLDPn (Modèle Lagrangien de Dispersion de Particules d'ordre n), which is a Lagrangian dispersion model developed at ECCC (D'Amours et al., 2015) which has been adapted for aquatic use (Paquin et al., 2020). Virtual trajectories are launched every 12 hours along the observed trajectory, between 20 September to 25



**Figure 5.** MAE contours (in km/day) between observed drift velocities with (3) for each of the four drifters. The left column uses CAPS forcing and the right column uses TOPAZ forcing. The black dot shows the location of the MAE minimum and the black contour line shows the MAE value within 1 km/day of the minimum. Each row is for an individual drifter in order from high ice concentration at the top to low ice concentration at the bottom. Sensitivity to choice of $\alpha_w$ is opposite to that of $\alpha_i$ in Figure 4 with less sensitivity in high ice concentration relative to low.

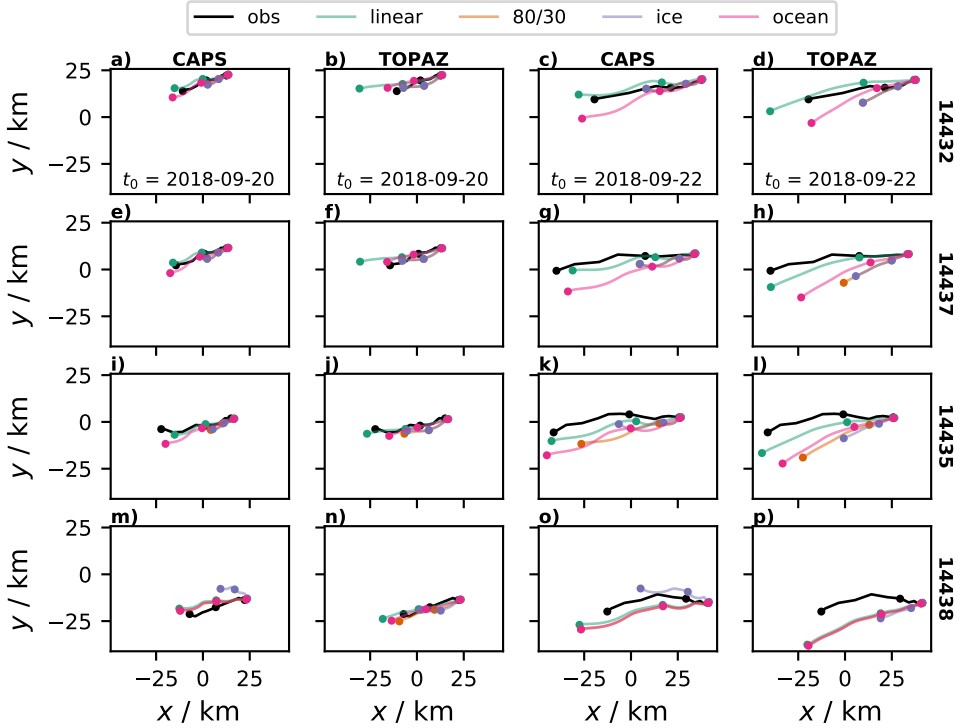

**Figure 6.** Examples of observed and modelled 48 hour trajectories. Each column corresponds to a combination of unique starting time and choice of ice-ocean forcing. Each row corresponds to a unique drifter. The dots represent locations at 24 hour intervals. The x and y axis are in km and the choice of origin is arbitrary. The different predicted tracks are linear using (3), 80/30 using (1), ice which uses only the ice velocities, and ocean which uses the ocean velocities plus 3% leeway.

September 2018. Two different drift models are used for the simulations. First, the classic drift equation given by (1) with $k_i^{80/30}$ and leeway coefficients $\alpha_w = 0.03$ and $\alpha_i = 0$. These leeway coefficients are chosen for the 80/30 method as they are typical values used in the MIZ (Nordam et al., 2019). Second, the more general (3) with the linear transfer function $k_i = A$ and $\alpha_w = 0.03$ and $\alpha_i = 0.02e^{-i\pi/6}$. For comparison, we test an ice-only transport ($k_i = 1$, $\alpha_i = 0$) and ocean-only transport ($k_i = 0$, $\alpha_w = 0.03$). The use of a leeway in the ocean is for consistency with 1).

A few examples of the observed and modelled trajectories for each drifter can be found in Figure 6. A more detailed presentation of the separation distance, $d$, after 48 hours between the observed and modelled trajectories as a function of trajectory start time is located in Figure 7. A summary of the mean and standard deviation of the separation distance $d$ can be found in Table 3).

For the drifter furthest in the ice, 14432, and the drifter second furthest in the ice, 14437, the separation distance after 48 hours is the smallest for forecasts beginning before 21 September and this does not vary much with choice of transport model or ice-ocean forcing (Figure 7a,b). This changes rapidly after 21 September where the predicted trajectories using the 80/30


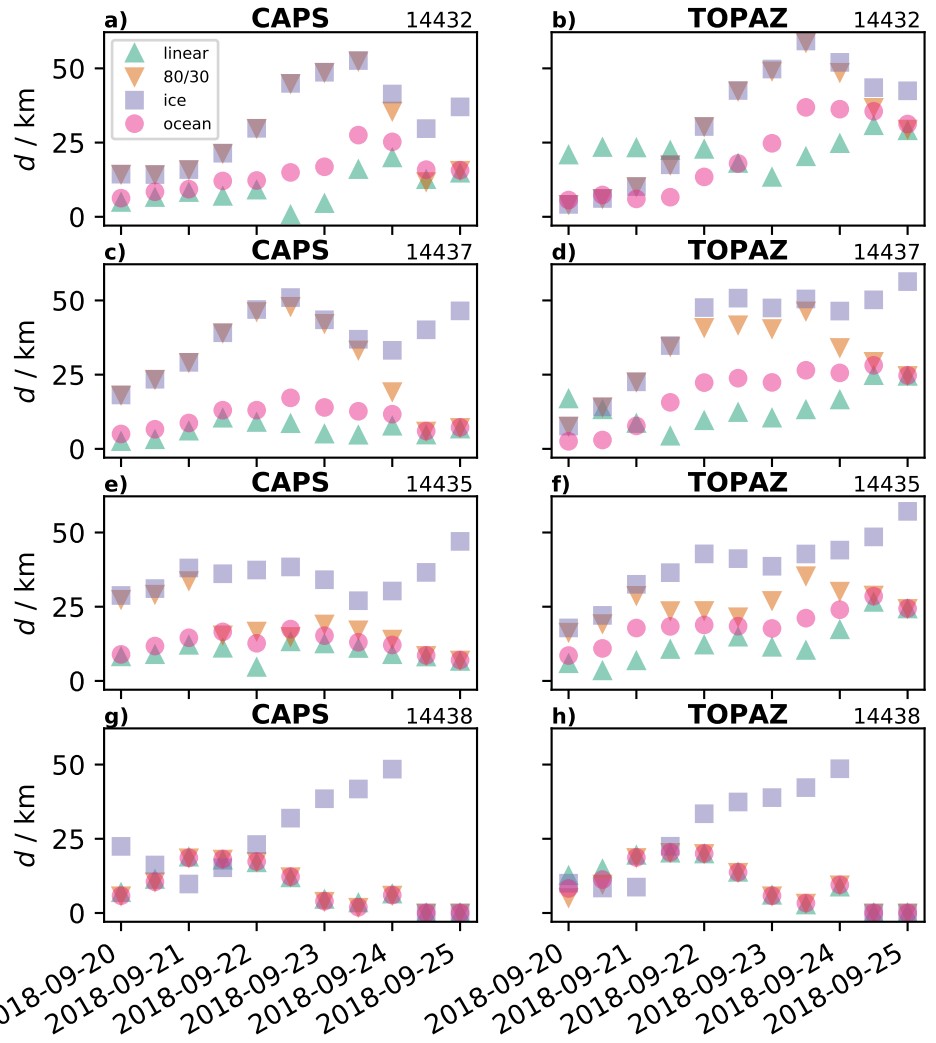

**Figure 7.** Drifter separation, $d$ in km, after 48 hours. The horizontal axis denotes the start times for each simulation simulation so that $d$ is measured at 48 hours after the time shown on the axis. The linear model corresponds to (3), the 80/30 model to (1), ice corresponds to the prediction with just ice velocities with no leeway and ocean corresponds to the prediction with just ocean velocities plus 3% leeway. The solid horizontal lines show the mean separation distance for each respective drift model. The inclusion of a leeway coefficient does reduce separation distance between observed and predicted trajectories with the linear weighted model providing the best results.





**Table 3.** Mean and standard deviation of separation distance, $d$, after 48 hours (in km) from Figure 7 for each drifter, forcing, and drift model.

| drift | 14432 | | 14437 | | 14435 | | 14438 | |
|-------|-------|------|-------|------|-------|------|-------|------|
| model | CAPS | TOPAZ | CAPS | TOPAZ | CAPS | TOPAZ | CAPS | TOPAZ |
| linear | $9.4 \pm 5.5$ | $18.2 \pm 8.9$ | $6.2 \pm 2.4$ | $15.0 \pm 7.0$ | $9.5 \pm 2.5$ | $14.4 \pm 6.9$ | $8.9 \pm 6.6$ | $10.1 \pm 7.0$ |
| 80/30 | $27.7 \pm 14.6$ | $30.4 \pm 17.9$ | $28.4 \pm 14.1$ | $30.6 \pm 11.7$ | $18.6 \pm 8.0$ | $25.4 \pm 5.1$ | $8.6 \pm 6.8$ | $9.7 \pm 7.3$ |
| ice | $31.8 \pm 13.5$ | $32.6 \pm 18.9$ | $37.1 \pm 9.8$ | $39.0 \pm 15.9$ | $35.0 \pm 5.4$ | $38.6 \pm 10.7$ | $22.5 \pm 15.6$ | $22.8 \pm 17.2$ |
| ocean | $15.0 \pm 6.3$ | $20.2 \pm 12.5$ | $10.5 \pm 3.8$ | $18.0 \pm 9.2$ | $12.5 \pm 3.2$ | $19.0 \pm 5.5$ | $8.6 \pm 6.8$ | $10.1 \pm 7.$ |

and ice-only transport models perform poorly (large separation distance), while the linear model and ocean-leeway models do not produce such a dramatic change (Figure 7a,b). The time when the transport models diverge corresponds with the increase in wind speed (Figure 3a). As the ice-only and 80/30 transport models do not have a leeway term (these two transport models are identical for $A > 0.8$), that the leeway term can improve 48 hour trajectories when using operational ice-ocean prediction systems.

Drifter 14435, which is in an intermediate ice concentration, has a slightly different response than the previous two drifters in higher ice concentration. Similar to the other drifters, the leeway models (ocean-leeway, linear) have smaller trajectory errors than the non-leeway models (ice-only, 80/30 for $A > 0.8$). Where drifter 14435 is different is that the intermediate ice concentrations associated with this drifter trajectory, approximately 60-80%, create two distributions where the separation distances are sometimes similar to the ice-only model (Figure 7e, before 21 September) and where the distributions are more similar to those of the ocean-leeway (Figure 7e, after 21 September).

For the drifter in the lowest ice concentration, 14438, the 80/30, linear, and ocean-leeway models all have essentially the same trajectory error. As the ice concentration, as provided by the model, are less than 0.3 (with the exception of TOPAZ before 21 September) it is expected that these three transport models should reproduce the same trajectory.

In general, the linear model using the best-fit leeway coefficients produces the smallest separation distance as well as the least amount of variation as a function of simulation start time (Table 3). The ice-only transport model had the largest trajectory errors even though we are using drifters on ice floes as our proxy for transport in the MIZ. The ocean-leeway had errors comparable to the linear model and the 80/30 model did not perform well in high ice concentration but did perform well in low ice concentration. These results were generally independent of the choice of CAPS or TOPAZ for the ice-ocean forcing.

## 5 Conclusions

Presented here is a general leeway model, which assumes a linear weighting of the ice and ocean velocities based on ice concentration and allows for a non-zero leeway in both the ice and ocean components and this leeway can vary in magnitude and direction. Optimal leeway coefficients are calculated by minimizing the error between observed drifter velocities in the MIZ and the model velocities calculated using ice-ocean velocities provided by two different coupled ice-ocean prediction





systems: CAPS and TOPAZ. This general leeway model is inspired by the leeway model used for oil transport in ice-covered
waters (Nordam et al., 2019), but we allow for an ice leeway to possibly not be zero to account for missing physics and
uncertainties in the ice model and assume a linear weighting for all ice concentrations and not only a subset.

By minimizing the error between the observed and modelled trajectories, optimal values for the water and ice leeway coef-
ficients, $\alpha_w$ and $\alpha_i$ respectively, were determined. Optimal values for $\alpha_w$ were found in the range $0.02 < \alpha_w < 0.04$ and for
$\alpha_i$ they were $0.01 < \alpha_i < 0.03$ with a mean direction of $30°$ to the right of the wind. Slightly larger values for $\alpha_i$ using the
TOPAZ data compared to the CAPS data. For $\alpha_w$, this range of values is consistent with 3% required for the prediction of
surface drifters (Sutherland et al., 2020) and oil spill modelling (Nordam et al., 2019). The $\alpha_i$ values are curiously consistent
with canonical values used for icebergs of about 2% of the wind and $30°$ to the right (Leppäranta, 2011).

To assess the quality of the general leeway model we compared the trajectory difference between several transport models
for a series of short-term (48 hour) forecasts using the two different ice-ocean prediction systems (CAPS and TOPAZ). The
general leeway model with the linear $k_i = A$ is used with the optimal leeway values of $\alpha_w = 0.03$ and $\alpha_i = 0.02$ and $30°$ to the
right of the wind direction. This is compared with the 80/30 transport equation used by the oil spill community, as well as an
ice-only transport ($k_i = 1$, $\alpha_i = 0$) as well as an ocean-leeway model ($k_i = 0$, $\alpha_w = 0.03$). Results were generally independent
of the choice of ice-ocean forcing. The linear general leeway model consistently had the smallest trajectory error for all four
of the drifters. Somewhat surprisingly the ocean-leeway model consistently had the second smallest errors, even for the drifter
in the highest ice concentration (14432). This is most likely due to the ice-ocean prediction systems being coupled so that the
respective velocities will be strongly correlated in the MIZ when the internal ice stresses are small. The inclusion of the leeway
is also key as it will compensate for missing physics, notably from surface waves which are not included in the ice-ocean
prediction systems, and can as well compensate for any biases in the respective drag coefficients in the atmosphere-ice-ocean
system. The 80/30 model had mixed results and generally had smaller prediction errors for smaller ice concentrations than for
large. This is most likely due to the lack of a leeway coefficient in the ice. The ice-only prediction had the largest errors, further
emphasizing the importance of a leeway coefficient for accurate short-term prediction of drift in the MIZ.

It is not obvious why the errors associated with calculating the leeway coefficients are similar between the CAPS and TOPAZ
forcing (Figures 4 and 5, while the separation errors are much smaller for CAPS than TOPAZ (Figure 7). One likely explanation
is due to the drifter observations being available every 3 hours, hence velocities calculated using forward differences are also
every 3 hours, while CAPS and TOPAZ provide currents every hour. While averaging the CAPS and TOPAZ forcing would
most likely reduce the magnitude of the MAE, it is not expected that this averaging would affect the relative sensitivity, i.e. the
optimal leeway coefficients should not change. As CAPS includes tides, and therefore has more variability at these frequencies
than TOPAZ (Figure 3), impacts on the magnitude of the MAE will probably be greater for CAPS than TOPAZ. However,
simulation of the trajectories don't require the observed drifter velocities and should therefore be more accurate, i.e. the results
in Figure 7 and summarized in Table 3.

It is clear from the available data that the inclusion of an ice leeway improves short-term predictions in the MIZ. However,
there is not enough information to ascertain whether this is due to missing physics in the prediction systems or correcting for
biases that are correlated with the wind. For the ice leeway, $\alpha_i$, it is not clear the physical origin. It is not uncommon to observe

wave motion in the MIZ (Rabault et al., 2020), which will result in the Stokes drift due to the wave motion in addition to "radiation stress" (Weber, 1987), which is the force due to the conservation of wave momentum that arises from the amplitude attenuation due to sea ice. This force can also affect push the ice edge and impact the thickness (Sutherland and Dumont, 2018), which will in turn affect how waves propagate into the MIZ (Sutherland et al., 2019). There are also large uncertainties associated with the drag coefficients (Heorton et al., 2019) that could also be part of the leeway coefficient. It should also be

emphasized that while the use of an ice leeway improved predictions in the MIZ, there is little reason to expect that this would be the case in the pack ice where internal stresses significantly impact the drift. In such a case, a more sophisticated expression for $\alpha_i$, which depends on ice concentration such that $\alpha_i \to 0$ as $A \to 1$ would most likely be required that also preserves some of the improvements shown in the MIZ.

*Data availability.* The drifter data can be accessed at https://thredds.met.no/thredds/catalog/metusers/jeanr/data_papers/drift_barents_2018/
catalog.html?dataset=metusers/jeanr/data_papers/drift_barents_2018/drift_data.nc. TOPAZ forecasts are available on the Copernicus Marine Enrivonment Monitoring Service FTP server (https://resources.marine.copernicus.eu) at nrt.cmems-du.eu/Core/ARCTIC_ANALYSIS_FORE CAST_PHYS_002_001_a/dataset-topaz4-arc-myoceanv2-be/. CAPS ice-ocean forecasts are available at http://dd.alpha.meteo.gc.ca/yopp/ model_riops/ and the atmospheric forecasts at http://dd.alpha.meteo.gc.ca/yopp/model_caps/.

*Author contributions.* This article was written by GS with input from all co-authors. GS conceived of the idea, with feedback from all co-
authors, and wrote the software for data analysis and production of figures. VA, LH and MD contributed to preliminary data analysis. JR designed the drifters and was responsible for data collection and quality control. JR, LH and ØB contributed to drifter deployment.

*Competing interests.* No competing interests are present.

*Acknowledgements.* GS and MD thank the Ocean Protection Plan (OPP) for funding and support of this work. JR, VA, LRH and ØB thank the Nansen Legacy (Arven etter Nansen) project for contributing to the development and deployment of the wave loggers. JR thanks the
Norwegian Research Council under the PETROMAKS2 scheme (project DOFI, grant number 28062) for funding. LRH and VA would also like to thank the Fram Centre MIKON Flagship program for support through the OSMICO project. Our warmest thanks go to the crew of the R/V Kronprins Haakon, for the time spent on the boat during the cruise and their help in deploying the drifters.





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
