# Peer review of "Estimating a mean transport velocity in the marginal ice zone using ice-ocean prediction systems"

_The Cryosphere, 2021_

## Referee Comment (RC2)

Review of the article titled "*Determining an optimal transport velocity in the marginal ice zone using operational ice-ocean prediction systems*" by Sutherland et al. (2021)

Recommendation: **Minor Revision**

**General Comments:**

Based on the oil transport equation in marginal ice zones (MIZs), the authors proposed a generalized transport equation for estimating the transport velocity in the MIZ by primarily introducing a leeway coefficient in the ice (i.e., $\alpha_i$) into the former equation. The transport velocity $u$, by design, then is a weighted mean of the ice and water velocities, either of which has been corrected by the respective leeway coefficient (i.e., $\alpha_i$ and $\alpha_w$). Using the field observations from 4 drifters, the authors further determined the optimal leeway coefficients ($\alpha_w = 0.03$ and $\alpha_i = 0.02e^{-\pi/6}$) which would minimize the MAE between the observations and the results predicted by the model.

I found the manuscript is very interesting and the general leeway model suggested here could be very useful for future operations in the Arctic. I therefore suggest to accept the manuscript once it goes through a **minor revision**. Please see my specific comments below.

**Specific Comments:**

L112: "as well as for wave models (Rogers et al., 2016)" to "... (Masson and Leblond, 1989; Rogers et al. 2016; Liu et al. 2020)"

Masson, D., & Leblond, P. (1989). Spectral evolution of wind-generated surface gravity waves in a dispersed ice field. Journal of Fluid Mechanics, 202, 43-81. doi:10.1017/S0022112089001096

Liu, Q., Rogers, W. E., Babanin, A., Li, J., & Guan, C. (2020). Spectral Modeling of Ice-Induced Wave Decay, Journal of Physical Oceanography, 50(6), 1583-1604.

L117: "... calculate their solutions " to "... calculate their source terms or source functions"?

L147-151: This paragraph does not read well. If I understood correctly, both the CAPS and TOPAZ simulations were forced by the CAPS winds. But line 148 presents that "TOPAZ is forced by ECMWF IFS ...". Please revise here for clarity.

L193: "... at a fixed value of $\alpha_w$" to "... $\alpha_w$ (0.03)"

L200: "... at a fixed value of $\alpha_i$" to "... $\alpha_i$ ($0.02e^{-i\pi/6}$)"

P12, Fig. 5 caption: "for each of the four drifters" to "... drifters with the constant $\alpha_i = 0.02e^{-i\pi/6}$"

- Fig. 2 uses the unit "m/s" for all the velocities. Figs. 4 and 5, however, adopt "km/day". I am a bit confused why two different units are used for velocities in these

figures. Furthermore, to better understand how large the errors are, it may be better to also include the relative error (i.e., in %) in Tables 1 and 2.

L206: "Lagrangian ... n), which is a ..." - delete "which is"

---

## Author Comment (AC1)

Thank you very much for your detailed and thoughtful comments. Below you will find our responses where the reviewer's comments are in *italics* followed by our responses in blue.

**Reviewer #1**

*This is an interesting paper on the adjustment necessary for the applied use of drift forecasts from short-range forecasting systems. However it is not possible to make a meaningful, statistically significant conclusion on its validity based on an extremely limited sample dataset of 4 drifter buoys, operating for a maximum of 2 weeks during Fall 2018. Given the authors are employed by the forecasting centers producing the model outputs it is possible to do a much more comprehensive analysis with the addition of drifter data from open sources such as International Arctic Buoy Programme (IABP). This will allow further testing to ensure that the results are valid both seasonally, and for varying compactness of the MIZ.*

*The abstract identifies that knowledge of drift transport in the MIZ is critical for applications including offshore operations and emergency response. It then states that the proposed approach can be used "for operational purposes in the MIZ". There is insufficient evidence presented in this paper to warrant this statement, and there is no attempt to explain or justify this in presenting the results or conclusions. It is also odd that in including this statement, that there is no attempt to verify this applicability through the operational monitoring sections of the Norwegian Meteorological Institute or Environment and Climate Change Canada with their Norwegian or Canadian Ice Services. The abstract attempts to link the approach to operational monitoring, however the term "operational" is used throughout in the limited definition of the research community in meaning only the routine production of data, not the quality assurance and support also included in operational monitoring services.*

*The recommendation is therefore for major revision, including a more thorough analysis with additional data sources.*

We would like to begin by clarifying that the study is more of a proof of concept than a comprehensive test of the drift model. The primary motivation of the study is to see how well two environmental prediction systems can predict the motion of buoys in the marginal ice zone. Current best practices for prediction in the MIZ performed poorly and we found that

if we used both the ice and ocean data in the MIZ that the predictions improved. There are not a lot of data available in the marginal ice zone (MIZ), which makes studies such as this important in assessing environmental prediction systems in the MIZ. Programs like the IABP are fantastic, but these buoys are deployed in pack ice and will only drift through the MIZ at times and locations depending on dynamics. Thus their presence in the MIZ is quite sparse. Using such data could be an interesting study, but not necessarily equivalent to buoys deployed in the MIZ. Also, determining whether the IABP buoys are in the MIZ is dependent on ice concentration data products and/or ice-ocean prediction systems, with each having their limitations (temporal and spatial resolution for data products and accuracy from ice-ocean prediction systems).

While we clearly do not have enough data to provide definitive values for the leeway parameters, we argue that we do have enough data to show that current best practices do not perform well in the MIZ for this experiment. We further argue that including a leeway term in the ice - the origin of which could be due to physics not included in the ice-ocean prediction system such as surface waves, or errors in the drag coefficients - reduces the errors with the available data. The scope of the paper is to show that a) best practices for predicting drift in the MIZ do not perform well and b) there are arguments for including a leeway coefficient in the ice. A systematic study to improve the values of the leeway coefficients is beyond this scope.

We should also clarify that we use "operational" as short-term prediction of environmental conditions in order to support operational activities. Examples of this are the knowledge of ocean currents and ice velocities required to support search and rescue or oil spill response. While the quality assurance and support are part of the operational services, these aspects are more part of system upgrades and general improvement and indirectly related to the accuracy of individual predictions. To avoid confusion, we have removed the term "operational" from most of the manuscript and replaced it with "short-term prediction in support of operational activities".

*As the transport model is dependent on sea ice concentration (SIC), it is heavily reliant on the accuracy of the source of this data and it's spatial resolution.*

Yes, the transport model is dependent on SIC, but this is more a product of the SIC dependence of the coupled prediction systems from which the ice and ocean velocities come from. These three parameters (SIC, ice velocity,

ocean velocity) are what we get from CAPS and TOPAZ and we assume they form the basis for the transport model. By using both the ice and ocean velocities in the transport model, the model attempts to compensate for SIC errors leading to inaccurate wind stress partitions in the ice-ocean prediction systems. The wind dependent term, i.e. the leeway, will inevitabily be more senstive to SIC, but current best practices used to predict drift in the MIZ assume no leeway in the ice and we feel we show quite clearly that one needs some leeway in the ice. This could be due to errors in SIC on the scales associated with the buoy motion, but also related to missings physics due to the absence of surface waves in the ice-ocean prediction systems as well as inaccuracies in drag coefficients to name a few. To explicitly show the sensitivity to SIC of our transport model, it is straightforward to rewrite Eq. (3) in the manuscript as

$$\mathbf{u}_o = \mathbf{u}_w + \alpha_w \mathbf{U}_{10} + k_i \left[ (\mathbf{u}_i - \mathbf{u}_w) + (\alpha_i - \alpha_w) \mathbf{U}_{10} \right],$$

which shows that the SIC concentration (assuming $k_i = \text{SIC}$) is only important for large differences between the ocean and ice components. Indeed if we set $k_i = 0$, equivalent to the ocean-only model in the manuscript, this performs quite well and much better than ice-only (no leeway), and the 80/30 transport model (except for SIC less than 30% where they are equivalent).

*The 80% threshold for assuming ice is or is not in free drift is based on observations, which cover much smaller areas than the typical 100 square kilometers of passive microwave (PMW) SIC products and 12.5 kilometer resolution of TOPAZ.*

The 80% threshold is based on the internal ice stress being negligible to the other forces at these ice concentrations for typical wind values and ice parameters. While it may be originally from observations, it is also parameterized into how the ice component of CAPS and TOPAZ calculate the stress. As the ice model has the same SIC input, the model dynamics should be approximately equivalent to a free drift model when the SIC is less than 80%. In CAPS and TOPAZ, the ice strength formulation from Hibler (1979), $P = P_* e^{-C_*(1-A)}$ and a value for $C_* = 20$ so it can be shown that $P(A = 0.8) = 0.02P(A = 1)$. The reviewer quite correctly points out the spatial scales of the SIC data products that contribute to the analysis are much larger, but these are related to the "constrained" scales of the model and the prediction systems will still make calculations on much smaller scales (grid resolution) which are necessary to support operational activities.

*P7 Figure 2 and L153: What is shown here is that both CAPS and TOPAZ fail to properly reproduce the MIZ in their SIC values, as a result proposing a drift correction weighted by SIC runs into an issue. This is due to assimilation of PMW SIC into both models that fails to properly represent the MIZ and ice edge, except if it is extremely compact. It would be interesting to see these 2 models compared against the openly available U.S. Naval Research Laboratory GOFS3.1 forecasts, where assimilation of SIC in the MIZ is augmented with use of the Multisensor Analyzed Sea Ice Extent " Northern Hemisphere (MASIE) product. Although CAPS gets its sea ice state from RIOPS/GIOPS, and it also assimilates ECCC Canadian Ice Service ice chart data which would provide better information on MIZ ice conditions, those ice charts only cover the Canadian Arctic area and not north of Svalbard, so the SIC data coming from CAPS in this study also originated in PMW SIC products. A more thorough analysis should be performed also including MIZ in the Canadian sector of the Arctic, e.g. Beaufort and Labrador Seas.*

Prediction systems in the Arctic will have errors in the SIC in the MIZ, which will impact the proportion of the wind stress going to the ice and to the water. This is part of the motivation of this study. How can we make best estimates of drift velocity using our prediction systems? We show that using information from both the ocean and ice model improves estimates of drift velocity in the highly coupled region of the MIZ. The primary aim of the paper is to demonstrate a proof of concept for improved drift estimates of the MIZ. A thorough comparison amongst the many products in the Arctic is beyond this scope.

*P5 L123 "various ice floes". Given ice type is important for understanding the drag coefficients and differences in drift behaviour, why is the stage of development of these floes and whether there was any deformation (ridging) present not recorded?*

The initial state of the ice floe was observed, but the evolution of the ice floes could not be recorded by the instruments. The ice floes at deployment were relatively flat with minimal deformation observed (CHECK WITH JEAN). This has been added to the text.

*P5 L144 "The horizontal resolution is about 12 km in the Arctic." No, it is exactly 12.5 km on the Polar Stereographic grid projection used.*

This has been clarified in the text.

*P6 Figure 1: "Contours of ice concentration". A contour is a line feature, what is shown is shaded discretized sea ice concentration.*

Corrected to "filled contour" as contours are calculated and the regions between the contours are filled with the colours corresponding to the colourbar.

*P15 Section 5 Conclusions. The proposed general leeway model needs a more comprehensive evaluation to warrant the conclusions here such as P16 L277 "It is clear from the available data that the inclusion of an ice leeway improves short-term predictions in the MIZ".*

We believe we state very clearly that from the available data that an ice-only prediction would be very different after 48 hours than a hybrid approach. As always, no research is definitive and more data and ideas will come forth and things evolve. The main point we aim to make is that the current methods for estimating drift velocity in the MIZ in support of operations such as search and rescue and/or oil spill response can give large errors over short time periods in the marginal ice zone. By using a weighted velocity of the model ice and ocean velocities the errors in this case are reduced. The applicability of these results across the Arctic remains to be seen, but for this particular case study they show to hold true.

*The text includes a few typographical and stylistic errors:*

*P2 L23, P2 L35, P2 L50, P3 L70: Replace "arctic" with "Arctic", as capitalization is used when referring to the geographic region.*

*P2 L27: Repetitive "typically".*

*P2 L40, P4 L109: Replace "don't" with "do not".*

*P2 L45: Replace "it's" with "it has"*

Thank you. We have corrected the above typographical and stylistic errors as suggested.

---

## Author Response (AR1)

Thank you very much for your detailed and thoughtful comments. Below you will find our responses where the reviewer's comments are in *italics* followed by our responses in blue.

**Reviewer #1**

*This is an interesting paper on the adjustment necessary for the applied use of drift forecasts from short-range forecasting systems. However it is not possible to make a meaningful, statistically significant conclusion on its validity based on an extremely limited sample dataset of 4 drifter buoys, operating for a maximum of 2 weeks during Fall 2018. Given the authors are employed by the forecasting centers producing the model outputs it is possible to do a much more comprehensive analysis with the addition of drifter data from open sources such as International Arctic Buoy Programme (IABP). This will allow further testing to ensure that the results are valid both seasonally, and for varying compactness of the MIZ.*

*The abstract identifies that knowledge of drift transport in the MIZ is critical for applications including offshore operations and emergency response. It then states that the proposed approach can be used "for operational purposes in the MIZ". There is insufficient evidence presented in this paper to warrant this statement, and there is no attempt to explain or justify this in presenting the results or conclusions. It is also odd that in including this statement, that there is no attempt to verify this applicability through the operational monitoring sections of the Norwegian Meteorological Institute or Environment and Climate Change Canada with their Norwegian or Canadian Ice Services. The abstract attempts to link the approach to operational monitoring, however the term "operational" is used throughout in the limited definition of the research community in meaning only the routine production of data, not the quality assurance and support also included in operational monitoring services.*

*The recommendation is therefore for major revision, including a more thorough analysis with additional data sources.*

We would like to begin by clarifying that the study is more of a proof of concept than a comprehensive test of the drift model for all regions and seasons. The primary motivation of the study is to see how well two environmental prediction systems can predict the motion of buoys in the marginal ice zone. Current best practices for prediction in the MIZ performed poorly and we

found that if we used both the ice and ocean data in the MIZ that the predictions improved. There are not a lot of data available in the marginal ice zone (MIZ), which makes studies such as this important in assessing environmental prediction systems in the MIZ.

Programs like the IABP are great, but using this data to predict drift in the MIZ is not trivial and would be significantly more work. In addition, most of the IABP data I found was daily or 12 hour time resolution with the exception of some data collected since 2021. For the Lagrangian analysis here, these coarser temporal resolution data will not resolve tides or any inertial oscillations which are important for short-term prediction. To us this really sounds like another study and not well suited for the short-time scale prediction that we are interested in.

There are other more minor items with using IABP drifters, such as that these buoys are deployed in pack ice and will only drift through the MIZ at times and locations depending on dynamics. Thus their presence in the MIZ will be localized to a few locations. Using such data could be an interesting study, but not necessarily equivalent to buoys deployed directly in the MIZ.

While there are surely uncertainties in the calculated leeway parameters, we argue that we do have enough data to show that current best practices do not perform well in the MIZ for this experiment. We further argue that including a leeway term in the ice - the origin of which could be due to physics not included in the ice-ocean prediction system such as surface waves, or errors in the drag coefficients - reduces the errors with the available data. The scope of the paper is to show that a) best practices for predicting drift in the MIZ do not perform well and b) there are arguments for including a leeway coefficient in the ice. A systematic study to improve the values of the leeway coefficients is beyond this scope.

We are not entirely clear with your last statement about our use of operational oceanography. The "routine production of data" as you phrase it is a key component of operational oceanography and related products. These products are used to support operational activities, which include incidents in the MIZ. This study shows that our approach performs better than standard approaches for the available data. Your comments about quality assurance and support appears to us to be more related to the ice-ocean prediction systems than a method to use their outputs. To avoid confusion we have clarified that this study aims to improve "short-term prediction in support of operational activities" and removed ambiguities associated with

operational oceanography.

*As the transport model is dependent on sea ice concentration (SIC), it is heavily reliant on the accuracy of the source of this data and it's spatial resolution.*

Any transport model (that we're aware of) in the MIZ is going to be dependent on SIC. This is part of the motivation of this study, to see if we can compensate for some of these uncertainties in SIC by using both ice and ocean velocities produced by coupled prediction systems. The wind dependent term, i.e. the leeway, will inevitabily be more senstive to SIC, but current best practices used to predict drift in the MIZ assume no leeway in the ice and we feel we show quite clearly that one needs some leeway in the ice. This could be due to errors in SIC on the scales associated with the buoy motion, but also related to missings physics due to the absence of surface waves in the ice-ocean prediction systems as well as inaccuracies in drag coefficients to name a few. To explicitly show the sensitivity to SIC of our transport model, it is straightforward to rewrite Eq. (3) in the manuscript as

$$\mathbf{u}_o = \mathbf{u}_w + \alpha_w \mathbf{U}_{10} + k_i \left[ (\mathbf{u}_i - \mathbf{u}_w) + (\alpha_i - \alpha_w) \mathbf{U}_{10} \right],$$

which shows that the SIC concentration (assuming $k_i = $ SIC) is only important for large differences between the ocean and ice components. Indeed if we set $k_i = 0$, equivalent to the ocean-only model in the manuscript, this performs quite well and much better than ice-only (no leeway), and the 80/30 transport model (except for SIC less than 30% where they are equivalent).

Another point in support of our lack of sensitivity to SIC is the similarity in prediction between CAPS and TOPAZ, which have different resolutions and SIC fields, which is consistent with the above points.

*The 80% threshold for assuming ice is or is not in free drift is based on observations, which cover much smaller areas than the typical 100 square kilometers of passive microwave (PMW) SIC products and 12.5 kilometer resolution of TOPAZ.*

Our understanding is that the 80% threshold is based on the internal ice stress being negligible to the other forces at these ice concentrations for typical wind values and ice parameters. While it may be originally from observations, it is also parameterized into how the ice component of CAPS and TOPAZ calculate the stress. As the ice model has the same SIC input,

the model dynamics should be approximately equivalent to a free drift model when the SIC is less than 80%. In CAPS and TOPAZ, the ice strength formulation is from Hibler (1979), $P = P_* e^{-C_*(1-A)}$ and a value for $C_* = 20$ so it can be shown that $P(A = 0.8) = 0.02P(A = 1)$. The reviewer quite correctly points out the spatial scales of the SIC data products that contribute to the analysis are much larger, but these are related to the "constrained" scales of the model and the prediction systems will still make calculations on much smaller scales (grid resolution) which are necessary to support operational activities.

*P7 Figure 2 and L153: What is shown here is that both CAPS and TOPAZ fail to properly reproduce the MIZ in their SIC values, as a result proposing a drift correction weighted by SIC runs into an issue. This is due to assimilation of PMW SIC into both models that fails to properly represent the MIZ and ice edge, except if it is extremely compact. It would be interesting to see these 2 models compared against the openly available U.S. Naval Research Laboratory GOFS3.1 forecasts, where assimilation of SIC in the MIZ is augmented with use of the Multisensor Analyzed Sea Ice Extent " Northern Hemisphere (MASIE) product. Although CAPS gets its sea ice state from RIOPS/GIOPS, and it also assimilates ECCC Canadian Ice Service ice chart data which would provide better information on MIZ ice conditions, those ice charts only cover the Canadian Arctic area and not north of Svalbard, so the SIC data coming from CAPS in this study also originated in PMW SIC products. A more thorough analysis should be performed also including MIZ in the Canadian sector of the Arctic, e.g. Beaufort and Labrador Seas.*

We do not disagree with your points about sea ice assimilation and the failure by CAPS and TOPAZ to reproduce the MIZ in their SIC values. However, as we argue earlier in our comment about SIC sensitivity, the aim of this study is to minimize the effects of these short comings on short-term drift prediction. By comparing CAPS and TOPAZ, which have different resolutions and SIC, we show the robustness of the method for two different ice-ocean models with different resolutions and model dynamics. Adding a third model, such as GOFS3.1, would be a lot more work and beyond the scope of the paper as we do not aim to suggest that this is the definitive drift model to be used in the MIZ. Rather, it is a proof of concept so the community at large can use and test in their respective configurations if such a method improves or degrades their drift prediction. We found a positive impact for our data, but more research is required.

*P5 L123 "various ice floes". Given ice type is important for understanding the drag coefficients and differences in drift behaviour, why is the stage of development of these floes and whether there was any deformation (ridging) present not recorded?*

The initial state of the ice floe was observed, but the evolution of the ice floes could not be recorded by the instruments. The ice floes at deployment were relatively flat with minimal deformation observed. This has been added to the text.

*P5 L144 "The horizontal resolution is about 12 km in the Arctic." No, it is exactly 12.5 km on the Polar Stereographic grid projection used.*

This has been clarified in the text.

*P6 Figure 1: "Contours of ice concentration". A contour is a line feature, what is shown is shaded discretized sea ice concentration.*

Corrected to "filled contour" as contours are calculated and the regions between the contours are filled with the colours corresponding to the colourbar.

*P15 Section 5 Conclusions. The proposed general leeway model needs a more comprehensive evaluation to warrant the conclusions here such as P16 L277 "It is clear from the available data that the inclusion of an ice leeway improves short-term predictions in the MIZ".*

We believe we state very clearly that from the available data that an ice-only prediction would be very different after 48 hours than a hybrid approach. As always, no research is definitive and more data and ideas will come forth and things evolve. The main point we aim to make is that the current methods for estimating drift velocity in the MIZ in support of operations such as search and rescue and/or oil spill response can give large errors over short time periods in the marginal ice zone. By using a weighted velocity of the model ice and ocean velocities the errors in this case are reduced. The applicability of these results across the Arctic remains to be seen, but for this particular case study they show to hold true.

*The text includes a few typographical and stylistic errors:*

*P2 L23, P2 L35, P2 L50, P3 L70: Replace "arctic" with "Arctic", as capitalization is used when referring to the geographic region.*

*P2 L27: Repetitive "typically".*

*P2 L40, P4 L109: Replace "don't" with "do not".*

*P2 L45: Replace "it's" with "it has"*

Thank you. We have corrected the above typographical and stylistic errors as suggested.

**Reviewer #2**

**General Comments:**

*Based on the oil transport equation in marginal ice zones (MIZs), the authors proposed a generalized transport equation for estimating the transport velocity in the MIZ by primarily introducing a leeway coefficient in the ice (i.e., $\alpha_i$) into the former equation. The transport velocity u, by design, then is a weighted mean of the ice and water velocities, either of which has been corrected by the respective leeway coefficient (i.e., $\alpha_i$ and $\alpha_w$). Using the field observations from 4 drifters, the authors further determined the optimal leeway coefficients ($\alpha_w = 0.03$ and $\alpha_i = 0.02e^{-i\pi/6}$) which would minimize the MAE between the observations and the results predicted by the model. I found the manuscript is very interesting and the general leeway model suggested here could be very useful for future operations in the Arctic. I therefore suggest to accept the "manuscript once it goes through a minor revision. Please see my specific comments below.*

Thank you for the excellent comments on the manuscript. We will address your specific comments individually below.

**Specific Comments:**

*L112: "as well as for wave models (Rogers et al., 2016)" to "... (Masson and Leblond, 1989; Rogers et al. 2016; Liu et al. 2020)"*

*Masson, D., & Leblond, P. (1989). Spectral evolution of wind-generated surface gravity waves in a dispersed ice field. Journal of Fluid Mechanics, 202, 43-81. doi:10.1017/S0022112089001096*

*Liu, Q., Rogers, W. E., Babanin, A., Li, J., & Guan, C. (2020). Spectral Modeling of Ice-Induced Wave Decay, Journal of Physical Oceanography, 50(6), 1583-1604.*

Added the additional references

*L117: "... calculate their solutions " to "... calculate their source terms or source functions?"*

This has been changed to emphasize the source terms are weighted by ice concentration, but only one solution for the wave action equation is calculated in the MIZ.

*L147-151: This paragraph does not read well. If I understood correctly, both the CAPS and TOPAZ simulations were forced by the CAPS winds. But line 148 presents that "TOPAZ is forced by ECMWF IFS ...". Please revise here for clarity.*

Yes, we can see how this is confusing. We have clarified in the text that only the CAPS winds are used in the leeway analysis.

*L193: "... at a fixed value of $\alpha_w$" to "... $\alpha_w$ (0.03)" L200: "... at a fixed value of $\alpha_i$" to "... $\alpha_i$ (0.02e^{-i\pi/6})"*

Thanks. The sentence has been reworded for clarity.

*P12, Fig. 5 caption: "for each of the four drifters" to "... drifters with the constant $\alpha_i = 0.02e^{-i\pi/6}$"*

Corrected to similar format as Fig. 4 caption.

*Fig. 2 uses the unit "m/s" for all the velocities. Figs. 4 and 5, however, adopt "km/day". I am a bit confused why two different units are used for velocities in these figures. Furthermore, to better understand how large the errors are, it may be better to also include the relative error (i.e., in %) in Tables 1 and 2.*

This is a good point (to clarify we know you meant Fig. 3). We show drifter velocities and model velocities in m/s as this is a typical choice for instantaneous values from these sources. We also output the errors in km/day as we are interested in the errors on the time scale of days plus km/day is a typical unit for ice drift. But you bring up a good point and we feel it could be useful to have both scales, which we have now added to Fig. 3.

Tables 1 and 2 show the mean average error, so what you are suggesting is to show the mean relative error (or more commonly the mean absolute

percentage error [MAPE])? This is an entirely different metric and would not simply be another column in Tables 1 and 2,

$$\text{MAPE} = 100 \left| \frac{\mathbf{u}_o - \mathbf{u}_m}{\mathbf{u}_o} \right|,$$

where $\mathbf{u}_o$ is the velocity of the object and $\mathbf{u}_m$ is the velocity of the model. This will create singularities when $\mathbf{u}_o$ is close to 0 (for example drifter 14438 around Sep-22). Also, we feel the units are easy to relate with the time-dependent analysis presented in Figs 6 and 7 and Table 3.

We calculated the MAPE and included the figures here. The MAPE is larger for the drifter with the smaller velocities (14432) as expected while the MAE is only slightly larger. We feel this new metric is not well suited for this study and opt to not include it.

*L206: "Lagrangian ... n), which is a ..." - delete "which is"*

Corrected.

[Figure]

Figure R1: Filled contours of MAPE (in %) between observed drift velocities and (3) for the along and cross-wind components of $\alpha_i$ with $\alpha_w = 0.03$. The left column uses the CAPS forcing and the right column uses TOPAZ forcing. The black dot shows the location of the MAE minimum and the black contour line shows the MAE value within 10% of the minimum. Each row is for an individual drifter in order from high ice concentration at the top to low ice concentration at the bottom. Sensitivity to to the choice of $\alpha_i$ is much greater in the high ice concentration than the low.

[Figure]

Figure R2: Filled contours of MAPE (in %) between observed drift velocities and (3) for the along and cross-wind components of $\alpha_i$ with $\alpha_w = 0.03$. The left column uses the CAPS forcing and the right column uses TOPAZ forcing. The black dot shows the location of the MAE minimum and the black contour line shows the MAE value within 10% of the minimum. Each row is for an individual drifter in order from high ice concentration at the top to low ice concentration at the bottom. Sensitivity to to the choice of $\alpha_i$ is much greater in the high ice concentration than the low.

---

## Author Response (AR2)

Thank you very much for your detailed and thoughtful comments. Below you will find our responses where the reviewer's comments are in *italics* followed by our responses in blue.

**Reviewer #1**

*This is an interesting paper on the adjustment necessary for the applied use of drift forecasts from short-range forecasting systems. However it is not possible to make a meaningful, statistically significant conclusion on its validity based on an extremely limited sample dataset of 4 drifter buoys, operating for a maximum of 2 weeks during Fall 2018. Given the authors are employed by the forecasting centers producing the model outputs it is possible to do a much more comprehensive analysis with the addition of drifter data from open sources such as International Arctic Buoy Programme (IABP). This will allow further testing to ensure that the results are valid both seasonally, and for varying compactness of the MIZ.*

*The abstract identifies that knowledge of drift transport in the MIZ is critical for applications including offshore operations and emergency response. It then states that the proposed approach can be used "for operational purposes in the MIZ". There is insufficient evidence presented in this paper to warrant this statement, and there is no attempt to explain or justify this in presenting the results or conclusions. It is also odd that in including this statement, that there is no attempt to verify this applicability through the operational monitoring sections of the Norwegian Meteorological Institute or Environment and Climate Change Canada with their Norwegian or Canadian Ice Services. The abstract attempts to link the approach to operational monitoring, however the term "operational" is used throughout in the limited definition of the research community in meaning only the routine production of data, not the quality assurance and support also included in operational monitoring services.*

*The recommendation is therefore for major revision, including a more thorough analysis with additional data sources.*

Allow us to clarify what we mean by "operational purposes in the MIZ". Our intent in using the term operational is twofold. First, we use it to imply the use of ice-ocean prediction systems that support operational activities related to drift in the MIZ and not the whole suite of operational services including ice monitoring. We will be sure to clarify this point in

the manuscript. The other is the need to predict single events using the best information available. As these buoys were purposefully deployed in the MIZ, they offered a case study for a single event in the MIZ to compare with results from two ice-ocean prediction systems that are used for operational purposes. Our results should be interpreted as such, i.e. a single case study in the MIZ, but we show that current best practices for drift prediction in the MIZ do not perform well for our case study and show some simple corrections reduce our error in short-term drift prediction.

Programs like the IABP are great, but using this data to predict drift in the MIZ is not trivial and would be significantly more work and in our opinion beyond the current scope of the paper. Using IABP data requires the determination of ice concentration in the MIZ, which is not trivial as we need to determine what dataset will be the definitive one for this. Also, to use the IABP data requires accessing all the available forecast data, determine if the buoy is in the MIZ, and then set up the drift experiment for each buoy. This is another study altogether and not a simple revision. We would also like to note that the temporal resolution of the IABP buoys we accessed is 12 hours (I did find higher frequency IABP buoys for 2021 but not before then), which is adequate to test trajectories but insufficient to obtain accurate velocity estimates to determine leeway coefficients.

We did check how many IABP drifters were available during the same time period as the drifter deployment as we already had this data easily available. There were a total of 264 IABP buoys in the Arctic during this period. To determine if they are in the MIZ we used the ice concentration from CAPS and interpolated this linearly in time and space to each IABP location. We deemed they were in the MIZ, and could be used for prediction, if their duration was at least 48 hours and at any point were in an ice concentration between 0.10 and 0.70 according to CAPS. These criteria leave us 21 IABP trajectories, which are shown in Figure R1a. As it takes time to set up drift experiments for each IABP buoy we select 94020 and 91680 as good candidates as they are relatively close together (limiting experimental setup) and appear on the edge of the pack ice and not coastal or polynas. I appreciate the font in Figure R1a is probably difficult to read, but these two are located on the top-left edge of the sea ice at approximately 140°W, 75°N.

To highlight some of the difficulties in determing if the drifter is in the MIZ we show the ice concentration from both CAPS and TOPAZ for each IABP buoy (Figure R1b). According to CAPS, both IABP drifters are only briefly in the MIZ with one mostly in high ice concentration and the other

[Figure]

[Figure]

Figure R1a: IABP trajectories, shown in red, that meet the MIZ criteria using CAPS as the ice concentration data. Drifter trajectories used in the study are shown in orange. Most of the IABP buoys are located in coastal regions and polynas (according to CAPS) with two trajectories, 94020 and 91680, being good candidates for testing. The shaded region shows the initial ice concentration from CAPS.

[Figure]

Figure R1b: Ice concentration ($A$) as a function of time for the two IABP buoys according to CAPS (left) and TOPAZ (right).

in predominantly low. TOPAZ data shows one drifter clearly in the MIZ (91680) but the other is in open water for the entire duration.

Part of our argument is that our method should be able to handle large uncertainties in ice concentration so we proceed with the same experiment and compare the synthetic 48 hour trajectories, with a trajectory started every 12 hours. An example of the trajectories for two start times is shown in Figure R1c and all the separation distances after 48 hours as a function of trajectory start time are presented in Figure R1d. We do not know if either of these IABP buoys are in the MIZ, but in general the linear model has smaller seperation distances for the buoy most likely in the ice (91680) and is still robust enough to perform well for buoy 94020 which is most likely in very low ice concentration.

In addition, most of the IABP data I found was daily or 12 hour time resolution with the exception of some data collected since 2021. For the Lagrangian analysis here, these coarser temporal resolution data will not resolve tides or any inertial oscillations which are important for short-term prediction. To us this really sounds like another study and not well suited for the short-time scale prediction that we are interested in.

There are other more minor items with using IABP drifters, such as that these buoys are deployed in pack ice and will only drift through the MIZ at

[Figure]

Figure R1c: Sample IABP trajectories and predictions for the different transport models and choice of ice-ocean forcing.

[Figure]

Figure R1d: Separation distance, in km, at 48 hour lead time as a function of start time for the two IABP buoys and different transport models and ice-ocean forcing.

times and locations depending on dynamics. Thus their presence in the MIZ will be localized to a few locations. Using such data could be an interesting study, but not necessarily equivalent to buoys deployed directly in the MIZ.

Given all the uncertainties with determining whether the buoy is located in the MIZ, we are hesitant to include this in the paper. In our opinion this is another study and not simply a revision of the current study. The scope of the current paper is to determine the accuracy of current operational prediction systems for a single event, which is highly relevant to the emergency response community, and if the method of Sutherland et al. (2021) could be applied to improve drift estimates. We believe we have enough evidence to show this and leave it to future work to investigate seasonal and regional variability of drift in the MIZ.

*As the transport model is dependent on sea ice concentration (SIC), it is heavily reliant on the accuracy of the source of this data and it's spatial resolution.*

Any transport model (that we're aware of) in the MIZ is going to be dependent on SIC. This is part of the motivation of this study, to see if we can compensate for some of these uncertainties in SIC by using both ice and ocean velocities produced by coupled prediction systems. The wind dependent term, i.e. the leeway, will inevitabily be more senstive to SIC, but current best practices used to predict drift in the MIZ assume no leeway in the ice and we feel we show quite clearly that one needs some leeway in the ice. This could be due to errors in SIC on the scales associated with the buoy motion, but also related to missings physics due to the absence of surface waves in the ice-ocean prediction systems as well as inaccuracies in drag coefficients to name a few. To explicitly show the sensitivity to SIC of our transport model, it is straightforward to rewrite Eq. (3) in the manuscript as

$$\mathbf{u}_o = \mathbf{u}_w + \alpha_w \mathbf{U}_{10} + k_i \left[ (\mathbf{u}_i - \mathbf{u}_w) + (\alpha_i - \alpha_w) \mathbf{U}_{10} \right],$$

which shows that the SIC concentration (assuming $k_i = \text{SIC}$) is only important for large differences between the ocean and ice components. Indeed if we set $k_i = 0$, equivalent to the ocean-only model in the manuscript, this performs quite well and much better than ice-only (no leeway), and the 80/30 transport model (except for SIC less than 30% where they are equivalent).

Another point in support of our lack of sensitivity to SIC is the similarity in prediction between CAPS and TOPAZ, which have different resolutions

and SIC fields, which is consistent with the above points.

*The 80% threshold for assuming ice is or is not in free drift is based on observations, which cover much smaller areas than the typical 100 square kilometers of passive microwave (PMW) SIC products and 12.5 kilometer resolution of TOPAZ.*

Our understanding is that the 80% threshold is based on the internal ice stress being negligible to the other forces at these ice concentrations for typical wind values and ice parameters. While it may be originally from observations, it is also parameterized into how the ice component of CAPS and TOPAZ calculate the stress. As the ice model has the same SIC input, the model dynamics should be approximately equivalent to a free drift model when the SIC is less than 80%. In CAPS and TOPAZ, the ice strength formulation is from Hibler (1979), $P = P_* e^{-C_*(1-A)}$ and a value for $C_* = 20$ so it can be shown that $P(A = 0.8) = 0.02P(A = 1)$. The reviewer quite correctly points out the spatial scales of the SIC data products that contribute to the analysis are much larger, but these are related to the "constrained" scales of the model and the prediction systems will still make calculations on much smaller scales (grid resolution) which are necessary to support operational activities.

*P7 Figure 2 and L153: What is shown here is that both CAPS and TOPAZ fail to properly reproduce the MIZ in their SIC values, as a result proposing a drift correction weighted by SIC runs into an issue. This is due to assimilation of PMW SIC into both models that fails to properly represent the MIZ and ice edge, except if it is extremely compact. It would be interesting to see these 2 models compared against the openly available U.S. Naval Research Laboratory GOFS3.1 forecasts, where assimilation of SIC in the MIZ is augmented with use of the Multisensor Analyzed Sea Ice Extent " Northern Hemisphere (MASIE) product. Although CAPS gets its sea ice state from RIOPS/GIOPS, and it also assimilates ECCC Canadian Ice Service ice chart data which would provide better information on MIZ ice conditions, those ice charts only cover the Canadian Arctic area and not north of Svalbard, so the SIC data coming from CAPS in this study also originated in PMW SIC products. A more thorough analysis should be performed also including MIZ in the Canadian sector of the Arctic, e.g. Beaufort and Labrador Seas.*

We agree with your points about sea ice assimilation and the failure by

CAPS and TOPAZ to reproduce the MIZ in their SIC values. However, as we argue earlier in our comment about SIC sensitivity, the aim of this study is to minimize the effects of these short comings on short-term drift prediction. By comparing CAPS and TOPAZ, which have different resolutions and SIC, we show the robustness of the method for two different ice-ocean models with different resolutions and model dynamics. Adding a third model, such as GOFS3.1, would be a lot more work and beyond the scope of the paper as we do not aim to suggest that this is the definitive drift model to be used in the MIZ. Rather, we present a case study so the community at large can use and test in their respective configurations if such a method improves or degrades their drift prediction. We found a positive impact for our data, but more research is required. With regards to your comment about a more thorough analysis including MIZ in the Canadian sector of the Arctic, we do not have such data and using IABP for MIZ studies introduces it's own difficulties (see earlier response) and beyond the scope of this case study on transport in the MIZ.

*P5 L123 "various ice floes". Given ice type is important for understanding the drag coefficients and differences in drift behaviour, why is the stage of development of these floes and whether there was any deformation (ridging) present not recorded?*

The initial state of the ice floe was observed, but the evolution of the ice floes could not be recorded by the instruments. The ice floes at deployment were relatively flat with minimal deformation observed. This has been added to the text.

*P5 L144 "The horizontal resolution is about 12 km in the Arctic." No, it is exactly 12.5 km on the Polar Stereographic grid projection used.*

This has been clarified in the text.

*P6 Figure 1: "Contours of ice concentration". A contour is a line feature, what is shown is shaded discretized sea ice concentration.*

Corrected to "filled contour" as contours are calculated and the regions between the contours are filled with the colours corresponding to the colourbar.

*P15 Section 5 Conclusions. The proposed general leeway model needs a more comprehensive evaluation to warrant the conclusions here such as P16 L277 "It is clear from the available data that the inclusion of an ice leeway*

*improves short-term predictions in the MIZ".*

We believe we state very clearly that from the available data that an ice-only prediction would be very different after 48 hours than a hybrid approach. We have softened the statement to be more matter of fact so now it reads as "The inclusion of a leeway coefficient in the ice reduces the prediction error for our drifter trajectories." As always, no research is definitive and more data and ideas will come forth and things evolve. The main point we aim to make is that the current methods for estimating drift velocity in the MIZ in support of operations such as search and rescue and/or oil spill response can give large errors over short time periods in the marginal ice zone. By using a weighted velocity of the model ice and ocean velocities the errors in this case are reduced. The applicability of these results across the Arctic remains to be seen, but for this particular case study they show to hold true.

*The text includes a few typographical and stylistic errors:*

*P2 L23, P2 L35, P2 L50, P3 L70: Replace "arctic" with "Arctic", as capitalization is used when referring to the geographic region.*

*P2 L27: Repetitive "typically".*

*P2 L40, P4 L109: Replace "don't" with "do not".*

*P2 L45: Replace "it's" with "it has"*

Thank you. We have corrected the above typographical and stylistic errors as suggested.

**Reviewer #2**

**General Comments:**

*Based on the oil transport equation in marginal ice zones (MIZs), the authors proposed a generalized transport equation for estimating the transport velocity in the MIZ by primarily introducing a leeway coefficient in the ice (i.e., $\alpha_i$) into the former equation. The transport velocity u, by design, then is a weighted mean of the ice and water velocities, either of which has been corrected by the respective leeway coefficient (i.e., $\alpha_i$ and $\alpha_w$). Using the field observations from 4 drifters, the authors further determined the optimal leeway coefficients ($\alpha_w = 0.03$ and $\alpha_i = 0.02e^{-i\pi/6}$) which would minimize the MAE between the observations and the results predicted by the*

*model. I found the manuscript is very interesting and the general leeway model suggested here could be very useful for future operations in the Arctic. I therefore suggest to accept the "manuscript once it goes through a minor revision. Please see my specific comments below.*

Thank you for the excellent comments on the manuscript. We will address your specific comments individually below.

**Specific Comments:**

*L112: "as well as for wave models (Rogers et al., 2016)" to "... (Masson and Leblond, 1989; Rogers et al. 2016; Liu et al. 2020)"*

*Masson, D., & Leblond, P. (1989). Spectral evolution of wind-generated surface gravity waves in a dispersed ice field. Journal of Fluid Mechanics, 202, 43-81. doi:10.1017/S0022112089001096*

*Liu, Q., Rogers, W. E., Babanin, A., Li, J., & Guan, C. (2020). Spectral Modeling of Ice-Induced Wave Decay, Journal of Physical Oceanography, 50(6), 1583-1604.*

Added the additional references

*L117: "... calculate their solutions " to "... calculate their source terms or source functions?"*

This has been changed to emphasize the source terms are weighted by ice concentration, but only one solution for the wave action equation is calculated in the MIZ.

*L147-151: This paragraph does not read well. If I understood correctly, both the CAPS and TOPAZ simulations were forced by the CAPS winds. But line 148 presents that "TOPAZ is forced by ECMWF IFS ...". Please revise here for clarity.*

Yes, we can see how this is confusing. We have clarified in the text that only the CAPS winds are used in the leeway analysis.

*L193: "... at a fixed value of $\alpha_w$" to "... $\alpha_w$ (0.03)" L200: "... at a fixed value of $\alpha_i$" to "... $\alpha_i$ (0.02$e^{-i\pi/6}$)"*

Thanks. The sentence has been reworded for clarity.

*P12, Fig. 5 caption: "for each of the four drifters" to "… drifters with the constant $\alpha_i = 0.02e^{-i\pi/6}$"*

Corrected to similar format as Fig. 4 caption.

*Fig. 2 uses the unit "m/s" for all the velocities. Figs. 4 and 5, however, adopt "km/day". I am a bit confused why two different units are used for velocities in these figures. Furthermore, to better understand how large the errors are, it may be better to also include the relative error (i.e., in %) in Tables 1 and 2.*

This is a good point (to clarify we know you meant Fig. 3). We show drifter velocities and model velocities in m/s as this is a typical choice for instantaneous values from these sources. We also output the errors in km/day as we are interested in the errors on the time scale of days plus km/day is a typical unit for ice drift. But you bring up a good point and we feel it could be useful to have both scales, which we have now added to Fig. 3.

Tables 1 and 2 show the mean average error, so what you are suggesting is to show the mean relative error (or more commonly the mean absolute percentage error [MAPE])? This is an entirely different metric and would not simply be another column in Tables 1 and 2,

$$\text{MAPE} = 100 \left| \frac{\mathbf{u}_o - \mathbf{u}_m}{\mathbf{u}_o} \right|,$$

where $\mathbf{u}_o$ is the velocity of the object and $\mathbf{u}_m$ is the velocity of the model. This will create singularities when $\mathbf{u}_o$ is close to 0 (for example drifter 14438 around Sep-22). Also, we feel the units are easy to relate with the time-dependent analysis presented in Figs 6 and 7 and Table 3.

We calculated the MAPE and included the figures here. The MAPE is larger for the drifter with the smaller velocities (14432) as expected while the MAE is only slightly larger. We feel this new metric is not well suited for this study and opt to not include it.

*L206: "Lagrangian … n), which is a …" - delete "which is"*

Corrected.

[Figure]

Figure R2a: Filled contours of MAPE (in %) between observed drift veloci-ties and (3) for the along and cross-wind components of $\alpha_i$ with $\alpha_w = 0.03$. The left column uses the CAPS forcing and the right column uses TOPAZ forcing. The black dot shows the location of the MAE minimum and the black contour line shows the MAE value within 10% of the minimum. Each row is for an individual drifter in order from high ice concentration at the top to low ice concentration at the bottom. Sensitivity to to the choice of $\alpha_i$ is much greater in the high ice concentration than the low.

[Figure]

Figure R2b: Filled contours of MAPE (in %) between observed drift velocities and (3) for the along and cross-wind components of $\alpha_i$ with $\alpha_w = 0.03$. The left column uses the CAPS forcing and the right column uses TOPAZ forcing. The black dot shows the location of the MAE minimum and the black contour line shows the MAE value within 10% of the minimum. Each row is for an individual drifter in order from high ice concentration at the top to low ice concentration at the bottom. Sensitivity to to the choice of $\alpha_i$ is much greater in the high ice concentration than the low.